# Systemic Actions of Breast Cancer Facilitate Functional Limitations

**DOI:** 10.3390/cancers12010194

**Published:** 2020-01-13

**Authors:** Ruizhong Wang, Harikrishna Nakshatri

**Affiliations:** 1Department of Surgery, Indiana University School of Medicine, Indianapolis, IN 46202, USA; rewang@iupui.edu; 2Department of Biochemistry and Molecular Biology, Indiana University School of Medicine, Indianapolis, IN 46202, USA; 3VA Roudebush Medical Center, Indianapolis, IN 46202, USA

**Keywords:** breast cancer, functional limitations, cachexia, skeletal muscle, microRNA

## Abstract

Breast cancer is a disease of a specific organ, but its effects are felt throughout the body. The systemic effects of breast cancer can lead to functional limitations in patients who suffer from muscle weakness, fatigue, pain, fibromyalgia, or many other dysfunctions, which hasten cancer-associated death. Mechanistic studies have identified quite a few molecular defects in skeletal muscles that are associated with functional limitations in breast cancer. These include circulating cytokines such as TNF-α, IL-1, IL-6, and TGF-β altering the levels or function of myogenic molecules including PAX7, MyoD, and microRNAs through transcriptional regulators such as NF-κB, STAT3, and SMADs. Molecular defects in breast cancer may also include reduced muscle mitochondrial content and increased extracellular matrix deposition leading to energy imbalance and skeletal muscle fibrosis. This review highlights recent evidence that breast cancer-associated molecular defects mechanistically contribute to functional limitations and further provides insights into therapeutic interventions in managing functional limitations, which in turn may help to improve quality of life in breast cancer patients.

## 1. Prevalence of Functional Limitations in Breast Cancer

Breast cancer is one of the most common malignancies affecting one in eight women in the United States. With disease progression, breast cancer causes dramatic systemic effects. Myopenia and cachexia under metastatic settings are observed in ~25% of breast cancer patients, particularly in women with triple-negative breast cancer [1,2]. Functional limitations are observed even before cancer diagnosis and are the major reasons for increased no-cancer related deaths in these patients [3,4]. In a recent study, a cohort of 2202 women with breast cancer was examined and approximately 40% breast cancer patients demonstrated at least one of the functional limitations [5]. Patients with functional limitations exhibit decreased volitional activity, whole body weakness, and fatigue [6,7,8,9,10]. The prevalence of functional limitations in breast cancer patients is dependent on disease stage, age, ethnicity, demographics, and physical compositions [5]. Indeed, functional limitations are more common among older breast cancer patients [5,11]. Women with functional limitations are more likely to be overweight or obese with less physically activity [5,12]. More critically, preclinical and clinical data demonstrate that functional limitations result in significantly shorter survival due to non-cancer causes of death [5,13].

## 2. Functional Limitation is Likely Due to Skeletal Muscle Dysfunction in Breast Cancer

Growing evidence points to skeletal muscle dysfunction as a major cause of cancer-associated functional limitations. Skeletal muscles represent 40%–50% of the total body mass of a healthy human, making them collectively one of the largest organ systems [14,15]. A defining characteristic of skeletal muscles is its remarkable capacity of generating power for motion of an individual to maintain activities and to ensure the quality of life [16,17]. Thus, muscle force production, fatigue, and twitch characteristics are functional properties that define skeletal muscle quality. Life quality change associated with aging, Duchenne muscular dystrophy (DMD), chronic obstructive pulmonary disease (COPD), and cancer is directly related to these muscle parameters [18,19,20,21,22]. Recently, specific decrements to skeletal muscle’s contractile quality have been identified in cachectic patients and tumor-bearing mice. Women with breast cancer are found to have reduced cross-sectional areas of a single muscle fiber compared with cancer-free controls. In addition, women with breast cancer have reduced fractional content of both subsarcolemmal and intermyofibrillar mitochondria [23]. This observation is consistent with a higher risk of dying due to low levels of muscle mass in women with breast cancer compared to women who have adequate muscle mass [24]. Furthermore, women with higher muscle mass demonstrate improved response to cancer treatment and overall survival [25,26]. Preclinical data from animal models of breast cancer demonstrated that mice with primary mammary tumors have reduced grip strength and declined motor activity [13]. In line with functional limitations, reduction of body fat and increased body free water were observed in the transgenic PyMT mouse model, which represents the luminal B subtype of human breast cancer. Tumor progression also caused decrement of grip strength and reduction of specific force generated by muscle contraction [27], which might be related to the decreased size of muscle fibers and muscle mitochondria [23]. Moreover, breast cancer-associated functional limitations, including muscle weakness and reduced muscle force, correlate with dynamic systemic changes in cytokine/chemokine levels and myogenic regulation (Figure 1). This process involves complex gene expression regulatory networks in the skeletal muscle differentiation hierarchy, which is influenced by the chemokines and cytokines (Figure 1). In breast cancer bone metastasis models, body weight loss is associated with skeletal muscle atrophy and specific changes in the activity of skeletal muscle ryanodine receptors and the calcium (Ca^2+^) release channel [27].

## 3. Altered Cytokines Contribute to Functional Limitations in Breast Cancer

Cytokines are a large family of polypeptides and small proteins, including interleukin (e.g., IL-1, IL-2, IL-4), interferon (IFN-α, β, γ), chemokine, and tumor necrosis factor (TNF) families. Cytokines produced by a variety of cell types exert their actions locally (autocrine and paracrine) or systemically by directly interacting with their specific membrane receptors [28,29]. It has been well established that cytokines secreted by host immune cells in response to tumors and/or by the tumor itself are associated with muscle wasting and functional limitations [30,31]. High plasma levels of IL-6, TNF-α, INF-γ, and IL-1β have been observed in both tumor-bearing animals and cachectic cancer patients [32,33]. Fatigued women who survive breast cancer have significantly higher serum levels of IL-1ra, sTNF-RII, and neopterin than non-fatigued women [34]. This cancer-associated increase in pro-inflammatory activity is common across many diseases including autoimmune, inflammatory, and infectious diseases [35,36,37,38]. In our preclinical models, we identified several cytokines/chemokines secreted by mammary tumor lines derived from PyMT+ and Neu+ mice. Both cell lines secreted Gm-csf, Tnf-α, Ccl-2, Ifn-γ, and IL-1α while Timp1 was uniquely secreted by the PyMT+ tumor line. Neu+ mammary tumor line secreted G-csf, Ccl-1, Ccl-5, Cxcl-1, Cxcl-2, Cxcl-10, and Il-1ra. Recently, we measured 33 circulating cytokines/chemokines in plasma of PyMT+ mice by using the Milliplex kit and reported that Tnf-α, Tgf-α, β, G-csf, Il-6, and Gm-csf are elevated, whereas Mip-2 Cxcl-2, Il-9, Cxcl-5, and Il-1α are decreased in tumor-bearing mice compared to age and sex-matched control animals [13]. Thus, the cancer genome has an influence on the type of cytokines/chemokines secreted by cancer cells and consequently on the type and severity of functional limitations. In addition, Peake and colleagues [29] reported that skeletal muscle itself expresses a variety of cytokines, including IFN-γ, TNF-α, LIF, TGF-β, and IL-6. Thus, cytokines produced locally and delivered systemically act on muscles through their specific receptors on myogenic cells. Indeed, myogenic cells express receptors that interact with cytokines including TNF-α, IL-1, IL-6, IFN-γ, and G-CSF [28,39,40].

Many cytokines have been well studied for their actions on cancer-associated muscle wasting and functional limitations. TNF-α directly binds to type 1 TNF-α receptor (TNFR1) on muscle cells resulting in increased reactive oxygen species production via mitochondrial electron transport and activation of NF-κB signaling pathways. NF-κB increases the activity of the ubiquitin/proteasome pathway, which accelerates the regulated degradation of muscle proteins and hence causes muscle weakness [41]. Indeed, our recent report showed that increased levels of Tnf-α, Tgf-β, and G-csf in plasma of PyMT+ mice were accompanied by lower muscle mitochondrial activity and muscle dysfunction, which were restored upon inhibition of NF-κB signaling pathways [13]. Interestingly, TNF-α and several other cytokines have been shown to modulate peripheral insulin sensitivity with a mechanism that affects the activation of the insulin receptor and downstream signaling molecules in the skeletal muscles. In addition, TNF-α downregulated insulin-like growth factor-1-dependent signaling pathways that reduced muscle anabolic capacity and enhanced pro-catabolic stimuli [32]. In animal models, treatment with Tgf-β results in muscle atrophy and fibrosis, which causes reduced muscle contraction force [40,42]. In breast cancer with bone metastasis, Tgf-β causes muscle weakness and reduces muscle contractibility [27,43]. Similarly, TGF-β and TNF-α enhance the expression of zinc transporter ZIP14, which blocks the differentiation of muscle stem cells, also called satellite cells (MuSCs), and enhances the loss of myosin heavy chain in the skeletal muscles [44]. Taken together, breast cancer-associated alterations in circulating cytokines contribute to tumor-associated muscle dysfunction (Figure 1). Cancer-specific genomic aberrations may determine the type of cytokines/chemokines produced by cancer cells and pathways affected in the skeletal muscles.

## 4. Impaired Myogenesis Contributes to Functional Limitations in Breast Cancer

Healthy skeletal muscles have self-renewal capacity via MuSCs to replace injured or dead myofibers [45,46]. Generally, MuSCs are located in niche reservoirs between muscle fibers and extracellular matrix (ECM) and maintained in a quiescent state through activities of several factors including MuSC-enriched transcription factor PAX7 [47,48]. When muscles are injured, MuSCs are activated as myoblasts expressing both PAX7 and MyoD. Myoblasts have proliferating ability to generate as many cells as needed to repair injury. While a portion of myoblasts expressing PAX7, but lacking MyoD, return to quiescent state, the other major portion expressing MyoD, but lacking PAX7, will go on to differentiate and form multinucleated myofibers, and finally replace injured muscle fibers physically and functionally by expressing various contractile proteins, such as myosin heavy chain (MyHC) and skeletal muscle alpha actin [47,49]. Tumors have a pronounced impact on the number of regenerating myofibers in tumor-bearing mdx mice, a popular model to study DMD, compared to the number of regenerating myofibers in tumor-free mdx mice [50]. Skeletal muscle in PDX mouse cancer models also shows reduced regenerative capacity due to dysregulation of Pax7. Pax7 is of importance during myogenesis at any age as Pax7 deficiency in mice results in cell cycle arrest and precocious differentiation of MuSCs, which leads to impaired muscle regeneration [51,52]. As reported previously, Pax7 not only contributes to MuSC activation and self-renewal but also regulates MuSC proliferation and differentiation by acting on its target genes such as Myf5 and MyoD [53,54]. Indeed, MyoD and Myf5 along with other myogenic regulatory factors Myf6 and MyoG, determine the specification of myogenesis and influence MyHC levels and metabolic properties of myofibers [55]. In several of these models, through the P38 MAPK signaling pathway, TNF-α negatively regulates Pax7 expression and contributes to impaired muscle regeneration [56]. In the PyMT+ mammary tumor model, tumor-bearing mice contained lower skeletal muscle Pax7 as well as lower levels of MyoD along with the upregulation of circulating Tnf-α [13]. As illustrated schematically in Figure 1, tumor-derived cytokines/chemokines can impair specific steps of the myogenic program and contribute to functional limitations [13].

In animals that lack inducible nitric oxide synthase (Nos2) expression, MuSCs fail to proliferate and differentiate as a muscle damage-induced increase of Pax7+ MyoD+ double-positive cells is significantly delayed in iNOS-deficient mice along with a significant decrease of MyoG compared to wild type mice [57], indicating that Nos2 is required for effective regeneration of muscle. Nos2 is an enzyme that regulates the synthesis of nitric oxide (NO), which is involved in adult skeletal muscle homeostasis, particularly by mediating muscle regeneration after injury [58,59,60,61]. NO promotes activation, fusion, and maintenance of the pool of MuSCs in acutely and chronically damaged muscles [62]. Skeletal muscle Nos2 expression in PyMT+ mammary tumor mice compared to wild type mice was significantly lower [13], which may have an impact on recovery from injury.

Both BCL-2 and caspase-3 are key regulators of cell apoptosis [63,64]. BCL-2 prevents programmed cell death by inhibiting shuttle proteins BAX and BAK, as enhanced activity of BAX and BAK leads to caspase-3 activation that results in cell death [65,66]. In this regard, in human disease of DMD, enhanced expression of caspase-3 is responsible for myofiber cell death and muscle atrophy [67]. However, a complex role of caspase 3 in skeletal muscle homeostasis is emerging [68]. Caspase 3 null mice compared to wild type mice have reduced muscle mass and myoblasts due to defects in differentiation to myofibers with a parallel downregulation of MyoD and MyoG. These findings are further confirmed in the C2C12 cell line by pharmacologically inhibiting caspase-3 activity. A recent in vivo study showed that inhibition of caspase-3 activity results in a profound disruption in skeletal muscle regeneration [69]. Taken together, caspase-3 is required for MuSC differentiation. With respect to the role of Bcl-2 in myogenesis, Bcl-2+ C2C12 cells co-express markers of early stages of myogenesis, including desmin, MyoD, and Myf-5, and expression of Bcl-2 promotes clonal expansion as the muscle colonies produced by cloned Bcl-2–null cells contain only about half as many cells as the colonies produced by cells with Bcl-2 [70]. Since cytokines such as TNF can influence the expression and/or activity of Bcl-2 and caspase 3, circulating chemokine/cytokine-mediated changes in pro-apoptotic and anti-apoptotic proteins in skeletal muscle could have a profound impact on skeletal muscle function in cancer patients.

Skeletal muscle expresses many miRNAs [71,72,73,74,75,76] that regulate MuSC proliferation, differentiation, and even muscle force production by modulating gene expression via transcriptional, post-transcriptional, epigenetic mechanisms, and nuclear genome organization [71,77,78,79,80]. Cancer affecting the expression of muscle-specific miRNAs has been reported recently in breast cancer [38], rhabdomyosarcoma [81], pancreatic and colorectal cancer [82], gastric cancer [83], and lung cancer [84]. We reported that circulating levels of cardiac and skeletal muscle-enriched miR-486 were lower in breast cancer patients with metastasis compared to healthy individuals [38]. In animal models of breast cancer, lower circulating miR-486 levels correlate with reduced miR-486 expression in skeletal muscle. Furthermore, proteins in the myogenesis (Pax7/MyoD/MyoG) and myofiber survival (Dock3/Pten/Akt) networks, which are regulated by and/or regulate miR-486, are deregulated in skeletal muscles of tumor-bearing transgenic mice compared to controls [38]. In mammary tumor-bearing animals, the downregulation of miR-486 was correlated with the upregulation of circulating Tnf-α, a cytokine that suppresses the expression of miR-486 in myoblasts in culture potentially via NF-κB signaling pathways [13,38]. Thus, miR-486 is an integral part of the myogenesis signaling network that involves Pax7, MyoD, myostatin, and NF-κB [15,85,86,87] and a potential target of cancer-induced chemokines/cytokines. Reduced skeletal muscle miR-486 expression is a major defect in DMD and transgenic expression of miR-486 in muscle could rescue muscular dystrophy phenotype in animal models [88]. In this regard, based on our preclinical studies, we had previously proposed that loss of miR-486 is responsible for functional limitations in breast cancer patients.

miR-206 is another one of the widely studied miRNAs in skeletal muscles [89]. miR-206 promotes myoblast differentiation of C2C12 cells by blocking cell cycle progression and by inducing myofiber formation [90] via its upstream regulator MyoD and downstream targets Pax7, Pax3, and estrogen receptor-alpha [91]. During myoblast differentiation, MyoD activates miR-206 expression that, in turn, downregulates Pax7 and Pax3 [92,93,94]. TGF-β and myostatin, which are upregulated in cancer patients, downregulate miR-206 and MyoD levels in skeletal muscle. Skeletal muscles of myostatin knockout mice compared to wild type mice have significantly higher levels of miR-206, while Tgf-β treatment reduces the expression of miR-206 in C2C12 myoblasts [95,96]. Local injection of miR-206 into injured skeletal muscles in rats enhanced muscle regeneration both morphologically and physiologically and effectively inhibited muscle fibrosis [97]. These observations indicate the role of miR-206 in muscle regeneration and muscle function. Thus, it is not surprising that elevated circulating Tgf-β in a PyMT+ mammary tumor model correlated with lower miR-206 in muscles and reduced muscle function (Figure 1) [13]. In addition, decreased circulating miR-206 is observed in breast cancer [98], renal cell carcinoma [99], melanoma [100], and osteosarcoma [101] patients.

## 5. Energetic Inefficiency in Skeletal Muscle Contributes to Functional Limitations in Breast Cancer

Cachexia, myopenia, and functional limitations in cancer patients are generally associated with a negative energy balance resulting from either reduced energy production or increased energy expenditure during disease progression [102,103,104]. Advanced cancer patients have enhanced thermogenesis that increases energy expenditure, which is consistent with the observation in cachectic tumor-bearing animals [105]. It is also known that impairment of the mitochondrial compartment results in decreased ATP production, leading to an energy deficit that becomes even worse since it is coupled with steadily increased energy expenditure. Consistent with this possibility, we have shown reduced muscle mitochondrial content in a PyMT+ mammary tumor model [13], and others have shown reduced ATP levels and increased activity of the energy sensor AMPK [106,107], implying that breast cancer decreases energy production in skeletal muscles. Thus, it is not surprising that skeletal muscles of PyMT+ animals contained lower levels of Atp2a1 [13], an energy transfer enzyme that regulates cellular calcium required for muscle contraction and muscle force production [108]. Interestingly, miR-486 and miR-206 are localized in mitochondria [109], and it is known that miR-486 protects the membrane potential of mitochondria to maintain cell integrity [110,111] through anti-apoptotic BCL-2 family proteins [112,113]. Thus, reduced miR-486 in skeletal muscles of tumor-bearing mice may affect mitochondrial activity that regulates energy production. Collectively, current literature suggests impaired energy production in skeletal muscle as a consequence of cancer, which may be responsible for functional limitation (Figure 1).

## 6. Altered Extracellular Matrix Contributes to Functional Limitations in Breast Cancer

Extracellular matrix (ECM) consists of a variety of collagens, elastins, fibronectins, tenascins, proteoglycans, and glycosaminoglycans [114]. ECM has dynamic influence on myofibers and muscle function through ECM-directed remodeling events [115,116]. Metalloproteinases (MMPs) play an essential role in ECM remodeling. MMP-2 and MMP-9 degrade collagen type IV of the basement membranes, but also activate focal adhesion kinase that may lead to changes in integrin function and modifications of the cytoskeleton [117]. During early disease stage of DMD, MMP-9, which is elevated due to NF-kB activity, enhances myogenesis. However, during the later disease stage of DMD, MMP-9 level is decreased, leading to the accumulation of fibroadipose tissues and reduced muscle strength [118]. These findings indicate the dual actions of MMP-9 in ECM remodeling and muscle regeneration. We reported that skeletal muscle of mammary tumor-bearing mice compared to wild type mice had elevated ECM deposition, which could be due to a reduction of Mmp-9, [13]. The specific role of Mmp-9 in skeletal muscle remodeling is evident from transgenic and knockout mouse models. Overexpression of Mmp-9 results in myofiber hypertrophy with increased fiber size and contractile force [119]. On the other hand, knockout of the Mmp-9 gene causes muscle atrophy by decreasing fiber cross-sectional areas and altering myofiber type distribution in mouse hindlimb skeletal muscles [120]. Thus, enhanced ECM and decreased Mmp-9 in the skeletal muscle may directly or indirectly contribute to muscle-associated functional limitations (Figure 1). Although MMP-9 has pro-tumorigenic and pro-metastatic roles, MMP-9 targeted therapies failed in clinical studies, potentially due to its requirement for proper skeletal muscle function [121]. MMP members other than MMP-9 could also play a role in skeletal muscle remodeling as the expression of laminin A, a key component of basal membrane in muscle and an Mmp-11 target, which is downregulated in muscles [122]. How cancer causes changes in skeletal muscle ECM is unknown but TGF-β is the likely culprit as TGF-β is implicated in multiple fibrotic diseases [123,124,125]. TGF-β functions to promote ECM preservation by enhancing collagen synthesis, expression of ECM and profibrotic genes, and inhibiting ECM degradation [126,127] through reduced MMP activity [128,129,130]. Interestingly, MMP-9 is a positive regulator of TGF-β as MMP-9 is required for the release of active TGF-β [131,132]. Therefore, in cancer, it is likely that circulating TGF-β rather than locally produced TGF-β contributes to enhanced ECM deposition in skeletal muscle.

## 7. STAT3 Signaling Contributes to Muscle Wasting in Cancer Cachexia

The role of STAT3 in muscle function has been reviewed [133,134]. It is reported that STAT3 contributes to muscle wasting in PDX animal models, as constitutively active STAT3 induces muscle fiber atrophy and exacerbates wasting in cancer cachexia [135,136] in an NF-kB dependent manner [137]. Conversely, inhibition of STAT3 reduces muscle atrophy in cancer [135,136]. These observations are consistent with STAT3 signaling negatively regulating MuSC expansion and skeletal muscle regeneration. Pharmacological inhibition of Stat3 activity enhances MuSC proliferation and muscle repair after injury or during diseased conditions [138,139]. Interestingly, a recent study demonstrated that Stat3 promotes the progression of MuSCs to myogenic lineage through mitochondrial respiration [140], indicating the involvement of mitochondria in muscle repair or muscle wasting in disease conditions. Indeed, JAK2/STAT3 signaling improves cardiac dysfunction by normalizing mitochondrial respiratory function in vivo and in vitro [141] through regulating reactive oxygen species (ROS) formation [142]. Mitochondria are one of the most important sources of ROS in skeletal muscles [143]. The role of ROS and redox signaling in muscle function and muscle regeneration has been reviewed [143,144,145]. Mitochondrial STAT3 has also been implicated in modulating mitochondrial DNA, mitochondrial transcription, and electron transport chain activity [146,147,148,149,150], further influencing cell survival and metabolism [151,152].

## 8. NF-κB is a Critical Signaling Relay Engaged in Breast Cancer Associated Functional Limitations

NF-κB represses myogenesis [153,154] and enhances muscle atrophy in various skeletal muscle diseases [155,156,157,158]. Activation of NF-κB in response to TNF-α blocks muscle differentiation by enhancing the degradation of the skeletal muscle-specific transcription factor MyoD [154,159]. Apart from TNF-α, increased TGF-β levels in circulation of breast cancer patients [160] can activate TAK1 [161,162], which phosphorylates IKK2 as well as NIK [163,164] and inhibits myogenesis by repressing the expression of MyoD [75,165].

It is noted that the exact role of the NF-κB signaling pathway depends on the state of the skeletal muscle disease and the nature of the disease. For instance, normal skeletal muscles have lower NF-κB activity with different transcriptional profiles compared to stressed muscles. During the late-stage of muscle differentiation, NF-κB acts as a negative regulator of differentiation through the transcriptional repressor YinYang1 [166]. During skeletal muscle atrophy, NF-κB induces murine ring finger-1 [155], an E3 ubiquitin ligase known to be involved in multiple models of skeletal muscle atrophy [167]. NF-κB represses MyoD by inducing its negative regulator PAX7 [50]. Recent studies have demonstrated that myostatin, another protein induced by NF-κB, represses miR-486 expression and inhibitors of this molecule can overcome aging-associated sarcopenia [87,168]. By contrast, miR-486 increases MyoD and its own expression by repressing Pax7 [169]. Furthermore, miR-486 itself controls NF-κB [86]. Mmp-9 regulates NF-κB activity, which further modulates Mmp-9 expression in skeletal muscles. NF-κB mediated induction of nitric oxide has been reported recently [170]. Therefore, there is an intricate signaling network with feed-forward and feedback loops comprising of NF-κB, PAX7, MyoD, myostatin, miR-486, and MMP-9 that regulate skeletal muscle homeostasis, which is disrupted in cancer (Figure 1).

In preclinical studies, activation of NF-κB within muscles results in severe muscle atrophy in mice [50,155]. Inhibition of NF-κB activity preserves muscle mass and reduces oxidative stress in transgenic and PDX tumor-bearing mice [171,172]. In response to denervation-associated muscle degeneration, IKK2 depletion prevents muscle atrophy, maintains fiber type, size, and strength, increases protein synthesis, and reduces protein degradation [156]. Treatment of tumor-bearing animals with a non-specific NF-κB inhibitor prevents tumor-associated muscle wasting via increased MyoD expression and inhibition of ubiquitin-conjugating enzyme E214K in skeletal muscles [173]. Thus, NF-κB and its associated signaling pathways are potential therapeutic targets to treat cancer-related muscle dysfunction (Figure 1). Indeed, we reported that the NF-κB inhibitor, dimethylaminoparthenolide (DMAPT), restored skeletal muscle functions including grip strength and motion performance in mammary tumor-bearing mice, accompanied by amelioration of few of the tumor-associated skeletal muscle molecular defects [13].

## 9. Perspective and Conclusions

Although there has been recent progress in understanding functional limitations in breast cancer, many areas may need to be further explored. The limited progress is mainly due to the perception that breast cancer patients, compared to other cancer patients, rarely experience cachexia. However, considering one in eight women in the United States is diagnosed with breast cancer and 25% of breast cancer patients, particularly those with triple-negative breast cancer, experience cachexia [2,174], additional studies are needed. Furthermore, myopenic obesity, which not only provides a misguided outlook but also impacts chemotherapy tolerability [26,175], is common in breast cancer patients. Lean muscle mass instead of BMI may need to be measured in breast cancer patients to ensure that there is no muscle loss. Mechanistic insights into myopenic obesity are needed to develop therapeutic intervention. In certain cases, adipose atrophy is also a concern as chemokines/cytokines that cause skeletal muscle dysfunction in cancer also mediate adipose atrophy [176]. Healthy interaction between skeletal muscle and adipocytes is essential as adipokines from adipocytes have a direct influence on myocytes, particularly with respect to metabolic pathways. Cancer-associated changes in neuromuscular and bone-skeletal muscle interaction are gaining attention. Recently, Baraldo and her colleagues reported that permanent deletion of the mTORC1 gene results in instability in the neuromuscular junction, which correlates with reduced treadmill performance [177]. Thus, it might be worthwhile to examine neuromuscular junction structure integrity and stability in skeletal muscles of breast cancer patients or animals with mammary tumors, as innervated skeletal muscle by tissue constructs has higher contractile forces compared to non-innervated skeletal muscle [178]. Gut microbiome is another upcoming area of research and it has recently been shown to affect skeletal muscle size, composition, and function via a proposed gut–muscle axis [179,180]. Since cancer-induced cytokines/chemokines as well as cancer treatments can affect the gut microbiome, it will be interesting to determine an association between the gut microbiome and muscle dysfunction in breast cancer patients. Consistent with this possibility, animals that lack a microbiota in gut have reduced muscle weight, reduced transcription of genes in mitochondria, and reduced neurotransmitter for neuromuscular junction [181]. Since breast cancer is not a single disease and broadly classified into five subtypes [182,183], how each subtype affects skeletal muscle function is unknown. Whether molecular defects are different among subtypes and disease stages of breast cancer remains unknown due to scarce clinical information and limited translational knowledge. Therefore, clinical management of cancer-associated functional limitations remains a great challenge for improving the life quality of cancer patients. In this regard, the majority of the anti-cachexia drugs tested in clinical trials so far have proven ineffective in improving muscle functional performance although a few of them had effects in increasing body weight and lean body mass [184,185].

Overall, our review outlines recent advances in breast cancer-induced functional limitations, highlighting the role of circulating cytokines/chemokines, microRNAs, mitochondria, ECM, and impaired myogenesis, as depicted in Figure 1. Molecular defects outlined here, as well as advances being made in the field of skeletal muscle biology, offer an opportunity to develop pharmacological interventions for the treatment of functional limitations in breast cancer in the near future.

## Figures and Tables

**Figure 1 cancers-12-00194-f001:**
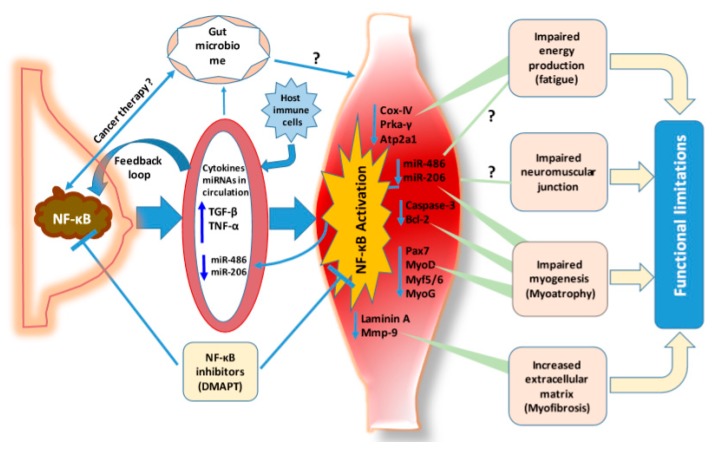
Systemic actions of breast cancer on skeletal muscles lead to functional limitations. Tumors in the breast release growth factors and cytokines into the circulating system. In response to tumors, host immune cells in breast tissue and other organs also release growth factors and cytokines into circulation. These growth factors and cytokines activate NF-κB in tumors, which further enhance the expression and release of growth factors and cytokines into circulation. Circulating growth factors and cytokines such as TGF-β and TNF-α are transported into skeletal muscles where, through their selective receptors, activate signaling molecules such as NF-κB in myogenic cells. Activated NF-κB regulates a series of molecules through its downstream signaling, feedback regulations, and cross-talk interactions. Tumor-induced reduction of Cox-IV, Prka-γ and Atp2a1 in skeletal muscles results in defective energetic regulation leading to fatigue in breast cancer patients. miR-486 and miR-206 are expressed in the mitochondria of myofibers, but their role in regulating energy remains unknown. Reduced expression of Pax7, MyoD, Myf5/6, and MyoG in skeletal muscles impairs proliferation and differentiation of MuSCs, which is exacerbated by downregulation of Bcl-2, miR-486, and miR-206. These actions lead to myogenic defects such as reduced number and size of myofibers, which is associated with myoatrophy. Downregulation of Mmp-9 and laminin A in skeletal muscles could contribute to increased ECM deposition and myofibrosis. Decreased expression of muscle-specific miR-486 and miR-206 may contribute to their lower levels in circulation. Whether the neuromuscular junction is affected by breast cancer is unknown. Furthermore, whether the tumor itself or cancer treatment affects the gut microbiome, which then changes muscle function, is unknown. Overall, breast cancer-induced impairment of myogenesis, energetic inefficiency, and ECM remodeling leads to functional limitations, which may be manifested clinically as fatigue, muscle weakness, fibromyalgia with and without overt cachexia. Pharmacological intervention through inhibitors of NF-κB such as DMAPT is a promising therapeutic strategy.

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
