# Peer review of "Systemic Actions of Breast Cancer Facilitate Functional Limitations"

_cancers, 2020, doi:10.3390/cancers12010194_

Round 1
Reviewer 1 Report
This manuscript reviews mechanisms of muscle dysfunction that occurs in the setting of breast cancer. The review is quite comprehensive and provides a large amount of data that will be useful to researchers and clinicians interested in this topic.
Specific Criticisms:
Most of my comments are related to style rather than content. The authors may consider the following edits:
Page 3, line 96: unclear the meaning of “f” after the word “interferon”Page 3, line 102: “survived from” should be replaced with the word “with”
Page 3, line 117: “including for” should be “including”
Page 3, line 119: “A quite a few of cytokines” should be “Quite a few cytokines” or just “Many cytokines”
The term muscle satellite cell (MuSC) is first introduced on page 4, line 134 but is clarified as stem cells in the next paragraph. It would be helpful to explain that they are stem cells when the term is first introduced.
Page 4, line 147, “onto” should be “on to”
Page 7, line 313 the use of the phrase, “few areas remain unexplored” implies that the topic has been extensively explored and this is opposite from what you are trying to say. Maybe consider saying that there are “many areas that need to explored”.
Author Response
This manuscript reviews mechanisms of muscle dysfunction that occurs in the setting of breast cancer. The review is quite comprehensive and provides a large amount of data that will be useful to researchers and clinicians interested in this topic.
Reply: We thank reviewer’s comments.
Specific Criticisms:
Most of my comments are related to style rather than content. The authors may consider the following edits:
Page 3, line 96: unclear the meaning of “f” after the word “interferon”
Reply: We thank the reviewer for pointing this out. We deleted “f”.
Page 3, line 102: “survived from” should be replaced with the word “with”
Reply: We thank the reviewer for pointing this out. We changed “from” to “with”.
Page 3, line 117: “including for” should be “including”
Reply: We thank the reviewer for pointing this out. We deleted “for”.
Page 3, line 119: “A quite a few of cytokines” should be “Quite a few cytokines” or just “Many cytokines”
Reply: We thank the reviewer for pointing this out. We changed it to “Many cytokines”.
The term muscle satellite cell (MuSC) is first introduced on page 4, line 134 but is clarified as stem cells in the next paragraph. It would be helpful to explain that they are stem cells when the term is first introduced.
Reply: We thank the reviewer for pointing this out. We clarified MuSC in the first introduction on page 4, line 134.
Page 4, line 147, “onto” should be “on to”
Reply: We divided “onto” to “on to”.
Page 7, line 313 the use of the phrase, “few areas remain unexplored” implies that the topic has been extensively explored and this is opposite from what you are trying to say. Maybe consider saying that there are “many areas that need to explored”.
Reply: We appreciate for reviewer’s comment and modified the text as“many areas may need to be further explored.”
Reviewer 2 Report
In the current review manuscript, authors have discussed the role of different factors involved in cancer-associated cachexia, with special emphasis on breast cancer-associated cachexia. The review is very concise and covered all the important aspects of functional limitations mainly muscular dysfunction caused due to breast cancer progression and metastasis. Authors have especially focused on myopathy, it will be great if they can shed some light on adipose atrophy. Discussion of other inflammatory pathways such as the role of STAT3 in breast cancer-induced myopathy will add value to the manuscript. Cancer-induced alteration in ROS homeostasis of skeletal muscle and its impact on myopathy should be discussed.
Author Response
In the current review manuscript, authors have discussed the role of different factors involved in cancer-associated cachexia, with special emphasis on breast cancer-associated cachexia. The review is very concise and covered all the important aspects of functional limitations mainly muscular dysfunction caused due to breast cancer progression and metastasis. Authors have especially focused on myopathy, it will be great if they can shed some light on adipose atrophy. Discussion of other inflammatory pathways such as the role of STAT3 in breast cancer-induced myopathy will add value to the manuscript. Cancer-induced alteration in ROS homeostasis of skeletal muscle and its impact on myopathy should be discussed.
Reply: We appreciate for reviewer’s comments and suggestions. We added a paragraph in the text to describing the role of STAT3 and ROS in skeletal muscle function and cancer. Adipose atrophy in the context of cachexia has been reviewed recently (Vegiopoulos et al., EMBO J 36:1999-2017) and we have included a short discussion based on this review in the Perspective and Conclusions section.